# Usefulness of P Wave Duration in Embolic Stroke of Undetermined Source

**DOI:** 10.3390/jcm9041134

**Published:** 2020-04-15

**Authors:** Moonki Jung, Jin-Seok Kim, Ju Hyeon Song, Jeong-Min Kim, Kwang-Yeol Park, Wang-Soo Lee, Sang Wook Kim, Gregory YH Lip, Seung Yong Shin

**Affiliations:** 1Cardiovascular & Arrhythmia Center, Chung-Ang University Hospital, Chung-Ang University, Seoul 06973, Korea; yessul88@naver.com (M.J.); jh1257@caumc.or.kr (J.H.S.); wslee1227@cau.ac.kr (W.-S.L.); swkimcv@cau.ac.kr (S.W.K.); 2Division of Cardiology, Department of Internal Medicine, Korea University Ansan Hospital, Ansan-si, Gyeonggi-do 15355, Korea; heartmania@unitel.co.kr; 3Department of Neurology, Chung-Ang University Hospital, Chung-Ang University, Seoul 06973, Korea; bellokim1@gmail.com (J.-M.K.); kwangyeol.park@gmail.com (K.-Y.P.); 4Liverpool Centre for Cardiovascular Science, University of Liverpool, Liverpool L69 3BX, UK

**Keywords:** subclinical atrial fibrillation, embolic stroke with undetermined source, P wave signal-averaged ECG

## Abstract

The investigation of the potential association between ischemic stroke and subclinical atrial fibrillation (SCAF) is important for secondary prevention. We aimed to determine whether SCAF can be predicted by atrial substrate measurement with P wave signal-averaged electrocardiography (SAECG). We recruited 125 consecutive patients with embolic stroke of undetermined source (ESUS) and 125 patients with paroxysmal atrial fibrillation as controls. All participants underwent P wave SAECG at baseline, and patients with ESUS were followed up with Holter monitoring and electrocardiography at baseline, 3, 6, and 12 months after discharge and every 6 months thereafter. In the ESUS group, 32 (25.6%) patients were diagnosed with SCAF during follow-up. There were no significant differences between the groups regarding atrial substrate. P wave duration (PWD) was a significant predictor of SCAF. Stroke recurrence occurred in 22 patients (17.6%), and prolonged PWD (≥ 135 ms) predicted stroke recurrence more robustly than SCAF detection. In ESUS patients, PWD can be a useful biomarker to predict SCAF and to identify patients who are more likely to have a recurrent embolic stroke associated with an atrial cardiopathy. Further research is needed for supporting the utility and applicability of PWD.

## 1. Introduction

It is essential to determine the underlying cause of an ischemic stroke to provide adequate secondary prevention. However, approximately 10–25% of ischemic strokes are still categorized as embolic strokes of undetermined source (ESUS) since they do not reveal any specific cause despite standard evaluation [1]. Prior analyses of patients with cardiac implantable electronic devices suggest that ESUS might be associated with subclinical atrial fibrillation (SCAF) and that continuous monitoring of electrocardiography (ECG) over a prolonged period is important to diagnose SCAF [2,3,4,5].

Nevertheless, it is difficult to use an implantable loop recorder (ILR) in all patients with ESUS; thus, efforts should be made to identify patients with a high likelihood of SCAF through other screening methods. More importantly, there is often no time correlation between the atrial fibrillation (AF) episodes identified via ILR and stroke events [2,6,7,8]. Indeed, patients with ESUS might not be submitted for appropriate secondary stroke prevention considering current screening and diagnosis approaches.

Generally, atrial arrhythmias can be initiated by ectopic beats (abnormal trigger) and further promoted by the atrial arrhythmogenic substrate, which refers to any structural or functional change in atrial myocardium. Both the abnormal trigger and atrial substrate are essential for the development of AF; however, the trigger and its consequential AF episode can only be recorded through a prolonged follow-up period. On the other hand, atrial substrate remodeling, characterized by inter- or intra-atrial conduction delay, is usually steady and can be reproducibly estimated regardless of the time of the examination.

P wave duration (PWD), assessed by signal-averaged electrocardiography (SAECG), is an established and precise method for evaluating inter- or intra-atrial conduction delay and has predictive value for AF under different circumstances [9,10,11]. In this study, we prospectively investigated the predictive value of PWD estimated by SAECG in predicting SCAF in patients with ESUS.

## 2. Materials and Methods

### 2.1. Study Population and Protocol

This study was a prospective, case-control, single-center trial designed to compare PWD estimated by SAECG between groups and to estimate the predictive value of PWD for SCAF in patients with ESUS. All consecutive patients diagnosed with ESUS from April 2015 to February 2018 at Chung-Ang University Hospital (Seoul, Korea) were enrolled in this study. Eligibility criteria included patients diagnosed with ischemic stroke with an embolic pattern on brain computed tomography or magnetic resonance imaging based on the TOAST (trial of ORG 10172 in acute stroke treatment) classification, demonstrating ischemic stroke > 1.5 mm in single or multiple vascular territories, or fragmented infarction. Patients with a history of mitral or aortic valve replacement, mitral stenosis, known AF, or cancer were excluded. Lacunar stroke was also excluded, and the lacunar stroke was defined as cortical infarction with a maximum dimension of ≤ 1.5 cm in the distribution of small, penetrating cerebral arteries.

The following tests were conducted to exclude major sources of cardioembolism and any other specific cause of stroke as well as to identify patients with undiagnosed AF—serial 12-lead ECGs during hospital stay; 24-hour Holter ECG monitoring; and 72-hour telemetry on the stroke unit with manual analysis. Transthoracic and transesophageal echocardiography were performed to confirm intracardiac thrombus, cardiac tumors, valvular vegetations, endocarditis, atrial septal defect, and patent foramen ovale. Transcranial Doppler ultrasonography and cervical duplex ultrasonography were performed to evaluate arteriovenous malformation and atherosclerotic plaques. Meanwhile, blood tests were performed to identify vasculitis or other inflammatory disease and hematologic conditions such as polycythemia and/or thrombocythemia. Patients with positive results in any of the aforementioned examinations were excluded from this study.

In this study, control subjects included patients with paroxysmal AF who agreed to undergo the P wave SAECG test. Control subjects with paroxysmal AF were excluded if they (1) had a low risk of stroke (CHA_2_DS_2_-VASc score 0-1), (2) were taking antiarrhythmic drugs, or (3) were in an AF state at the time of the P wave SAECG test. This study was approved by the Institutional Review Board of Chung-Ang University Hospital, and all participants provided informed consent.

### 2.2. P Wave Signal-Averaged Electrocardiography (SAECG)

A MAC 5500 HD system (Version 10B, GE Healthcare system, Milwaukee, WI, USA), with three orthogonal bipolar leads, X, Y, and Z, and high resolution P wave analysis algorithm (Phi-Res, GE Healthcare system, Milwaukee, WI, USA) was used for data acquisition and analysis. Silver/silver chloride electrodes were placed in an orthogonal lead arrangement. The anode and cathode of the X axis were placed at the left and right midaxillary line of the fourth intercostal space, respectively. The anode of the Y axis was attached to the left iliac crest, while the cathode of the Y axis was placed at the superior aspect of the manubrium of the sternum. The anode of the Z axis was placed at the V2 position, and the cathode of the Z axis was placed on the patient’s back, directly posterior to the anode of Z electrode. A ground electrode was placed on the right eighth rib. A sinus P wave template was selected for averaging. The QRS complex in ECG was used as the trigger for the signal averaging process. The signal was digitalized at a frequency of 16,000 samples/sec/channel with 16-bit accuracy. P wave complexes that did not match the template with a 99% correlation coefficient were automatically rejected until a noise level < 0.3 μV was reached. The individual X, Y, and Z leads were combined into a vector sum by the formula (X^2^ + Y^2^ + Z^2^)^1/2^, and a least-square-fit filter was applied to the averaged output. The least-square-fit filter was used since it decreased signal distortion and ringing of the P wave [11,12]. The following measurements were analyzed: unfiltered PWD, P wave vector integrals, and root-mean-square voltage for the terminal 20, 30, and 40 ms.

### 2.3. Follow-Up

ESUS patients underwent a P wave SAECG test prior to discharge in a stabilized state after exiting the stroke unit. Follow-up visits were scheduled at 3, 6, and 12 months after discharge, and every 6 months thereafter. All the follow-ups included a clinical examination, a 12-lead ECG, and a 24-hours Holter ECG. All ECGs were analyzed and adjudicated. If a patient complained of AF-related symptoms, we recommended that the patient visit the clinic immediately, regardless of the scheduled follow-up visits, and ECG documentation was attempted. A clinical diagnosis of AF was made in cases presenting an episode of AF lasting longer than 30 s.

### 2.4. Statistical Analysis

Normally distributed continuous variables were expressed as means and standard deviations, and categorical data were expressed as numbers and percentages. Nonparametrically distributed data were reported as median values with interquartile ranges. For comparison across groups, continuous variables were compared using the Student’s *t*-test or analysis of variance, as appropriate, and categorical variables were analyzed using the χ^2^ test or Fisher’s exact test, as appropriate. A multivariate logistic regression analysis was performed to determine the predictors of recurrent stroke. All of the potential confounders on the basis of the clinical significance or with a *p* value < 0.1 in the univariate analyses, were included in a stepwise regression analysis as the variable-selection process. Receiver operating characteristic (ROC) analysis was performed to assess the predictive performance of PWD. Kaplan–Meier curve and log rank test were performed to determine the effect of subclinical AF detection by PWD. A *p* value of < 0.05 was considered statistically significant. All statistical analyses were performed using SPSS 22 (IBM Corp., Armonk, NY, USA) and STATA version 13.0 (Stata Corp, College station, TX, USA).

## 3. Results

Between April 2015 and February 2018, a total of 125 consecutive patients were diagnosed with ESUS and enrolled in the study (69 male, 56 female, mean age 68.4 ± 12.1 years), while 125 patients with paroxysmal AF were recruited as a control group (68 male, 57 female, mean age 65.3 ± 12.2 years). All participants underwent P wave SAECG at baseline (Table 1). There was no significant difference in clinical data between the two groups except for age and recurrent stroke rate. The incidence of recurrent stroke was higher in ESUS group than in the control group (17.6% vs. 1.6%, *p* < 0.001). The normal value of PWD was set at 135 ± 7 ms based on Hofmann et al. [12]. There was no difference when comparing the normal range of PWD of the previous study with that of our young and non-AF subjects, although the data of this comparison is not presented in this article. Patients were followed up for a mean 1.6 ± 0.8 years. All the P wave SAECG tests were performed within 20–30 min and were well tolerated without any significant adverse events. There was no significant difference between the two groups in terms of slow conduction and pathological electrical remodeling of the atria, including PWD (134.5 ± 15.4 vs. 132.4 ± 17.5 ms, *p* = 0.321).

During follow-up, SCAF was detected in 12-lead ECG and 24-hour Holter ECG for 32 in 125 patients with ESUS (25.6%): 22 (17.6%) subjects were first diagnosed with SCAF by the Holter ECG and 10 (8%) subjects were first diagnosed with SCAF by the 12-lead ECG. The mean time to SCAF detection after an index ESUS event was 120.9 ± 163.7 days. The event rate for SCAF detection during the follow-up period was 20.6 events per 100 person-years in the ESUS group.

### 3.1. Performance of PWD in Clinical Detection of SCAF

The PWD assessed by P wave SAECG was effective in identifying patients with SCAF among patients with ESUS (Figure 1, C-index of standard PWD = 0.657, 95% CI 0.552 – 0.761, *p* = 0.008). When Yuden’s method was applied, the detection of SCAF was best predicted by PWD ≥ 135 ms, with sensitivity of 65.6% and specificity of 65.3%. For PWD ≥ 135 ms, the event rate of SCAF detection was 36.3 events per 100 person-years, whereas for PWD < 135 ms, the rate was 11.3 events per 100 person-years. 

To assess the risk factors associated with AF detection, logistic regression analysis showed that PWD (≥ 135 ms) was associated with an increased risk of AF (*p* value < 0.05, OR 3.883, 95% CI 1.331–11.327, Table A1). Kaplan–Meier curves comparing the two groups (PWD ≥ 135 ms vs. PWD < 135 ms) demonstrated significantly increased SCAF detection in patients with prolonged PWD (≥ 135 ms) (Figure 2, log rank test *p* = 0.002).

### 3.2. Recurrent Stroke during Follow-Up

A total of 22 recurrent stroke events were identified during the follow-up period despite maintaining antiplatelet therapy (APT) for secondary prevention of stroke. The mean time to recurrent stroke after an index ESUS event was 252.0 ± 106.7 days. The detailed distribution of patients with recurrent stroke based on the SCAF detection and the PWD prolongation (≥ 135 ms) is shown in Figure 3.

Only one-third of recurrent strokes occurred in patients identified with SCAF, and the remaining two-thirds occurred in patients in which SCAF was not detected; however, there was no statistically significant difference in stroke recurrence rate between SCAF and non-SCAF groups (8/32 (25%) vs. 14/93 (15.1%), *p* = 0.203). Among 14 patients with recurrent stroke in the non-SCAF group, 8 patients had a baseline PWD ≥ 135 ms (atrial cardiopathy (AC) group). The incidence of recurrent stroke in the union of SCAF and AC groups was higher than in the remaining non-SCAF and non-AC group (16/65 (24.6%) vs. 6/60 (10.0%), *p* = 0.037; Figure 3). 

Furthermore, only two patients (25%) experienced recurrent stroke with SCAF and were changed from APT to oral anticoagulant (OAC) therapy before stroke recurrence, while five (62.5 %) were treated with APT at the time of stroke recurrence. In terms of prevention strategy according to the medication used in this study, although patients on OACs seemed to have a lower incidence of stroke than those on APT, the difference was not statistically significant (7.7% vs. 20.2%, *p* = 0.161; Figure 4). Logistic regression analysis demonstrated that PWD prolongation was a more significant predictor of stroke recurrence (odds ratio (OR) 2.756) than SCAF detection (OR 1.881, not significant; Table 2).

## 4. Discussion

This study provides several important findings: (1) atrial electrical remodeling substrate assessed by P wave SAECG in patients with ESUS was equivalent to that of patients with clinically diagnosed paroxysmal AF; (2) we could predict SCAF through PWD assessed by SAECG, which is associated with atrial substrate, and consequently with the degree of intra-/interatrial conduction delay; (3) the standardized clinical follow-up allowed us to detect SCAF in approximately one-quarter of patients with ESUS, which is similar to the AF detection rate reported in a 1-year follow-up study using ILR in patients with ESUS [5]; (4) prolonged PWD (≥ 135 ms) predicts stroke recurrence more robustly than SCAF detection; (5) considering the stroke recurrence trend shown in this study, the current APT-based strategy is inadequate in preventing stroke recurrence in patients with ESUS (Figure 4).

### 4.1. PWD as an Alternative Screening Target for SCAF

Because of the previously reported association between ESUS and SCAF, detection of SCAF has been used as a screening target [4,13,14]. Although various strategies and modalities for prolonged, continuous ECG monitoring have been attempted to improve SCAF detection efficacy, their results have been inconsistent and often controversial [4,15,16,17,18,19]. 

Many investigators have tried to improve SCAF detection efficacy, for example, by investigating the impact of the AF burden or the potential influence of the ECG definition of AF [12,13]. However, these potential factors are largely dependent on the trigger, which has huge variations over time and is difficult to predict and document. Alternatively, we can try to predict the likelihood of SCAF by evaluating the atrial substrate, which constitutes another important factor that progressively contributes to the development of AF. In this study, we adopted the PWD, as estimated by P wave SAECG (an examination tool that has already proven to be predictive of AF in different conditions [9,10,11]), as a method for obtaining a precise and reproducible measurement of existing atrial substrate. In terms of atrial substrate assessed by SAECG, patients with ESUS were found to be comparable to patients with paroxysmal AF. Among patients with ESUS, PWD by SAECG was found to have a moderately predictive power in detecting SCAF, as was found in previous studies [9,11]. According to the Cryptogenic Stroke/ESUS International Working Group, the rate of AF detection varies between 2.7 and 30%, depending on duration and modality of monitoring. In the results of this study, 125 patients were followed for 1.6 ± 0.8 years, and 32 (25.6%) were diagnosed with SCAF. This is in line with the results presented by the Cryptogenic Stroke/ESUS International Working Group.

### 4.2. Clinical Significance of ESUS Patients without Clinical AF 

Only one-third of patients with advanced atrial substrate, represented by PWD ≥ 135 ms, were found to have SCAF, despite careful follow-up. Furthermore, most SCAF detection occurred within the first year of follow-up, and no further detection occurred thereafter in patients with advanced atrial substrate (PWD ≥ 135 ms); in patients with less atrial substrate (PWD < 135 ms), SCAF detection increased after two years of follow-up and a tendency to approach the curve of patients with advanced atrial substrate was observed (Figure 2). 

Assuming that the SCAF detection rate through ILR is the maximum value for our current scheme [5], our result can be interpreted in two ways. First, approximately one-third of patients with advanced atrial substrate (PWD ≥ 135 ms) may be clinically diagnosed with AF within one year after ESUS. The remaining two-thirds may not have the clinical presentation of AF but may have AC characteristics associated with ESUS [19,20]. In this study, although patients with less atrial substrate (PWD < 135 ms) have fewer SCAF detection in the earlier period after an ESUS event, some patients may have potential atrial substrate for SCAF. Since this cut-off value (PWD ≥ 135 ms) is determined arbitrarily to predict SCAF, it may differ from that of the AC we are seeking to identify. To clarify the boundary of AC, future studies should look for a cut-off value of PWD to determine AC but not SCAF. Second, such a detection pattern might be partly explained by frequent follow-ups within one year after the first ESUS event. However, given the similar SCAF detection rate with another study using ILR [5], the SCAF detection pattern may not be due to the frequency of the follow-ups or detection modality; instead, it may be due to the distinctive clinical manifestations, the so-called lone AC. 

Nevertheless, it is difficult to determine the specific criteria for AC based on the results of the present study; however, only approximately one-third of ESUS patients combined with AC may develop clinical AF, while the remaining two-thirds may have a similar clinical risk of stroke without the presentation of clinical AF. In terms of secondary prevention, it may be more practical to differentiate subjects with AC than to try to detect SCAF subjects with a low likelihood of SCAF detection. 

### 4.3. ESUS, Recurrent Stroke, and PWD

Interestingly, recurrent stroke was more frequent in patients with ESUS (17.6%) than in those with PAF (1.6%, *p* < 0.001), despite the similar demographic characteristics in both groups. For this point, some explanations may be possible as follows: first, the degree of thromboembolic risk in ESUS patients might differ from that in PAF patients. Indeed, diabetes mellitus is a frequent comorbidity factor in patients with AF and increases the risk of stroke by alterations of autonomic function [20]. In addition, DM causes a higher rate of subclinical AF events and strokes even in patients with lower CHA_2_DS_2_-VASc scores [21]. Furthermore, the chronicity of DM is known to be independently associated with ischemic stroke [22]. In this regard, although there was no difference in the proportion of diabetic patients between ESUS and PAF groups in our study, the duration of DM may be longer in patients with ESUS than in those with PAF; second, patients with ESUS may be more likely to have an atrial arrhythmogenic substrate causing AF and further recurrent embolic stroke. In this setting, AF recurrence could be related to the expression and regulation of specific families of microRNAs [23], to the expression and regulation of proteins implied in genesis and perpetuation of electro-anatomical reentry as proinflammatory molecules [24], and to the arrhythmic regulators of ionic currents [25]. To date, the overexpression of proinflammatory molecules and proteins implied in the regulation of ionic currents through calcium handling could cause a higher rate of AF events, greater persistence of AF, and increased refractory to catheter ablation treatments [24,25]. The present study demonstrated that prolonged PWD (> 135 ms) could be a predictor for recurrent stroke in patients with ESUS, suggesting that PWD can be useful to identify patients at high risk for stroke.

### 4.4. Secondary Prevention 

OAC was expected to be superior to APT in most patients with ESUS at the time that ESUS was defined; however, NAVIGATE ESUS, the global randomized controlled trial failed to show that rivaroxaban is superior to aspirin with regard to the prevention of recurrent stroke after an initial ESUS [26,27]. Although the results of the on-going randomized controlled trials will be required before definitive conclusions can be made [28,29], a subgroup analysis of the Warfarin-Aspirin Recurrent Stroke Study (WARSS) trial using warfarin in combination with a biomarker (N terminal pro-B-type natriuretic peptide, NT-proBNP > 750 pg/mL) [29,30] provides an early insight for the optimal strategy to prevent recurrent embolic stroke. The original WARSS trial comparing warfarin and aspirin in all patients with ischemic stroke and without considering the accompanying cardiac conditions other than AF, failed to demonstrate the superiority of warfarin [30,31]; however, in this subanalysis using a biomarker that reflects the accompanying cardiac condition, the use of warfarin in patients with NT-proBNP > 750 pg/mL showed 70% relative risk reduction compared with aspirin (*p* = 0.021) [29,30].

Similarly, the present study also failed to show the superiority of OAC to aspirin in the secondary prevention of stroke for ESUS patients (Figure 4). However, when examining stroke recurrence patterns in this study, the different results may be expected (Figure 3). Due to insufficient evidence, APT was mostly used as a primary agent for secondary prevention before SCAF detection regardless of the amount of atrial substrate at baseline. With this secondary prevention strategy, 17.6% of patients with ESUS experienced recurrent stroke during a mean follow-up of 1.6 years. When these patients were reclassified according to the prescribed drug at the time of stroke recurrence, we might observe that the stroke recurrence was reduced to one-third in patients prescribed OAC compared with patients prescribed APT. Furthermore, PWD could significantly predict stroke recurrence in patients with ESUS (Table 2). Therefore, considering the stroke recurrence trend shown in the present study, the current APT-based strategy is inadequate in preventing stroke recurrence in patients with ESUS. Further, our results suggest that the PWD can be a useful biomarker to identify patients who can benefit from OAC therapy. 

### 4.5. PWD as an Extended and Convenient Screening Target for AC

In a recent pooled analysis of patients with ESUS, APT was used as a secondary prevention method in 87%, with a substantial stroke recurrence rate of 4.5%/year [31,32]. Considering the limited preventive efficacy of APT in patients with ESUS, we should stratify patients who can benefit from changing APT to OAC. The addition of biomarkers measured through imaging, blood, or ECG can give us useful information, and several biomarkers have proven efficacious in the screening of patients that could potentially benefit from OAC [33,34,35]. 

Among various biomarkers, PWD assessed by SAECG is a noninvasive, reproducible, quantitative, and direct measure of atrial substrate for SCAF and AC. PWD can be assessed within 20–30 min and requires no additional follow-up time. Furthermore, it is possible to identify patients with AC who have not been differentiated from patients with ESUS by current SCAF tracking schemes and who can potentially benefit from using OAC. The expected benefit of OAC in these patients with AC among patients with ESUS should at least be investigated in future randomized controlled trials.

### 4.6. Limitations

This prospective analysis presents some limitations. First, a main limitation could be the absence of continuous monitoring techniques such as ILR or telemonitoring systems to detect AF events in the study population [36]. Indeed, these techniques could detect AF recurrence more effectively and avoid the associated negative prognosis with detection failure. Second, our study had relatively short follow-up periods and a small number of patients with a recurrent stroke event. However, these limitations did not negate the ability of PWD to predict both SCAF and stroke recurrence in patients with ESUS. Finally, we defined the clinical AF as episodes of AF that lasted > 30 s. Although this complies with the accepted definition of AF in general, there have been different definitions of AF in many clinical studies. A recent meta-analysis reported that SCAF strongly predicts clinical AF and is associated with elevated absolute stroke risk [37]. Thus, PWD ≥ 135 ms without SCAF detection may be related to AC with or without AF of shorter duration (< 30 s) or of a very rare frequency, thereby explaining the predictive value of PWD for recurrent stroke. Further research is needed to refine the association between PWD and stroke. 

## 5. Conclusions

PWD, an ECG biomarker associated with atrial substrate, is useful for predicting SCAF in patients with ESUS. Furthermore, advanced atrial substrate represented by PWD prolongation may be a more robust predictor of stroke recurrence than SCAF detection. Further research is needed for supporting the utility and applicability of PWD.

## Figures and Tables

**Figure 1 jcm-09-01134-f001:**
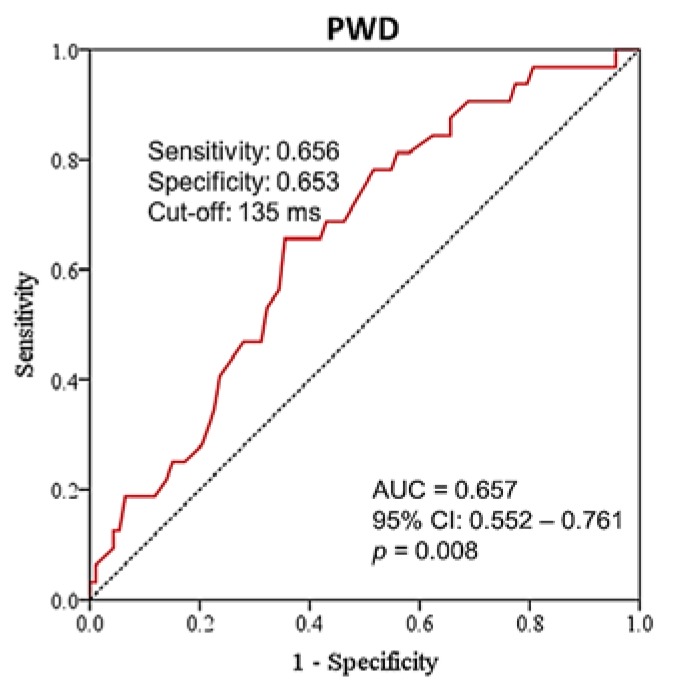
Receiver operating characteristic curve (ROC) of the signal-averaged P wave duration for predicting subclinical atrial fibrillation in patients with embolic stroke of undetermined source. AUC = area under the ROC curve, CI = confidence interval, PWD = P wave duration.

**Figure 2 jcm-09-01134-f002:**
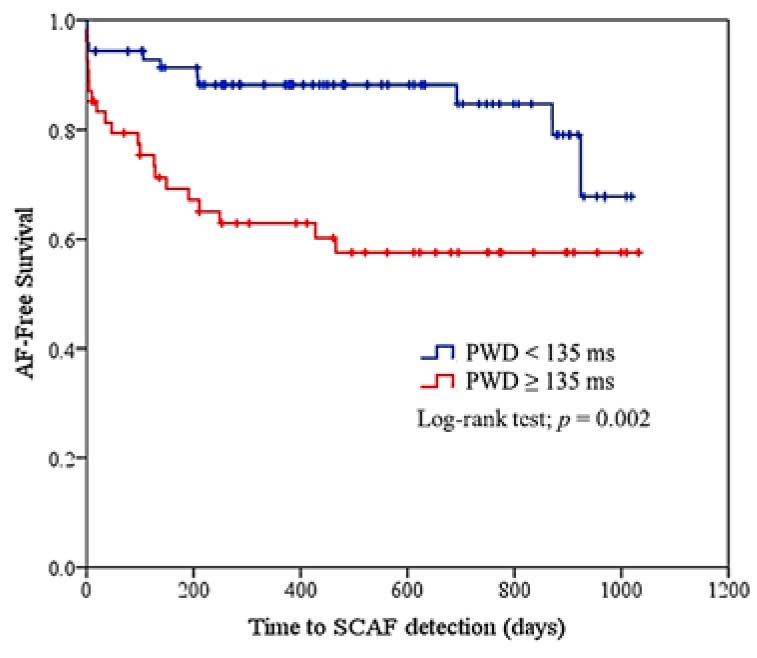
Kaplan–Meier curve for atrial fibrillation free survival by P wave duration levels. PWD (≥ 135 ms) was associated with an increased risk of AF (OR 3.883, 95% CI 1.331–11.327). AF = atrial fibrillation, PWD = P wave duration, SCAF = subclinical atrial fibrillation.

**Figure 3 jcm-09-01134-f003:**
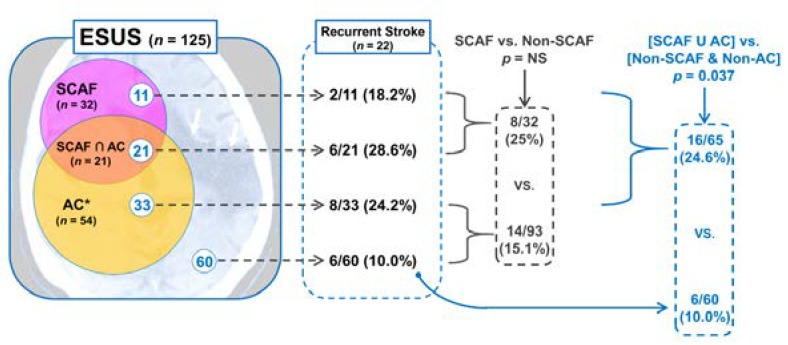
Venn diagram that summarizes causes of ESUS and detailed distribution of recurrent stroke. AC = atrial cardiopathy, ESUS = embolic stroke of undetermined source, NS = not significant, SCAF = subclinical atrial fibrillation. *: atrial cardiopathy was defined as P wave duration (PWD) ≥ 135 ms; “∩” and “U” symbols indicate an intersection and a union of two groups, respectively.

**Figure 4 jcm-09-01134-f004:**
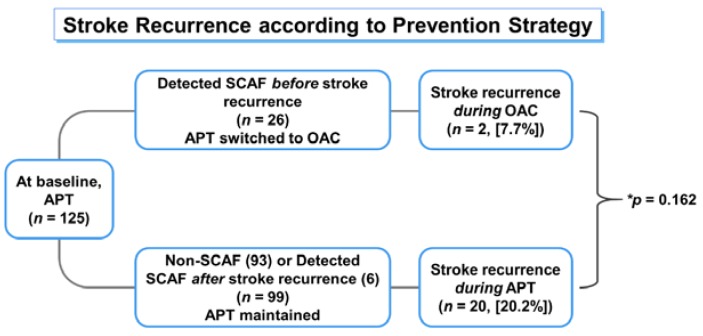
Stroke recurrence according to secondary prevention strategy. APT = antiplatelet therapy, OAC = oral anticoagulant, SCAF = subclinical atrial fibrillation. *: comparison of stroke recurrences between patients with OAC use and patients with APT use.

**Table 1 jcm-09-01134-t001:** Baseline characteristics and results of P wave signal-averaged ECG.

Variables	PAF (*n* = 125)	ESUS (*n* = 125)	*p* Value *	ESUS with SCAF (*n* = 32)	ESUS wo SCAF (*n* = 93)	*p* Value †
Age (years)	65.3 ± 12.2	68.4 ± 12.1	0.045	70.9 ± 7.8	67.6 ± 13.2	0.089
Male (%)	68 (54.4)	69 (55.2)	1	15 (46.9)	54 (58.1)	0.272
BMI (kg/m^2^)	24.0 ± 3.7	24.7 ± 4.0	0.090	25.2 ± 3.4	23.5 ± 3.7	0.290
CHA_2_DS_2_-VASc score	2.4 ± 1.5	2.7 ± 1.5	0.282	2.9 ± 1.5	2.6 ± 1.5	0.330
HTN (%)	80 (64.0)	87 (69.6)	0.420	25 (78.1)	62 (66.7)	0.224
DM (%)	24 (19.2)	22 (17.6)	0.871	3 (9.4)	19 (20.4)	0.188
Previous stroke or TIA (%)	10 (8.0)	7 (5.6)	0.451	4 (12.5)	3 (3.2)	0.070
Recurrent stroke (%)	2 (1.6)	22 (17.6)	<0.001	8 (25.0)	14 (15.1)	0.280
New onset AF (%)	0 (0)	32 (25.6)	<0.001	32 (100)	0 (0)	<0.001
**Signal-averaged ECG**						
Standard PWD (ms)	134.5 ± 15.4	132.4 ± 17.5	0.321	139.4 ± 44.2	131.1 ± 27.3	0.212
Total PWD (ms)	138.1 ± 26.2	137.3 ± 31.0	0.839	143.1 ± 42.6	135.3 ± 25.8	0.221
Terminal 40 ms (μV)	4.6 ± 2.9	4.3 ± 2.5	0.506	4.5 ± 3.1	4.2 ± 2.4	0.603
Terminal 30 ms (μV)	3.9 ± 2.6	3.5 ± 2.2	0.151	3.7 ± 2.5	3.4 ± 2.1	0.549
Terminal 20 ms (μV)	3.2 ± 2.5	2.7 ± 1.9	0.116	2.7 ± 2.2	2.7 ± 1.8	0.997
RMS voltage of P wave (μV)	6.3 ± 2.3	6.1 ± 2.2	0.580	6.2 ± 2.9	6.1 ± 2.0	0.855
Integral of P wave (μV·ms)	628.0 ± 240.1	609.5 ± 225.3	0.531	615.2 ± 277.4	607.6 ± 206.1	0.888
Noise (μV)	0.38 ± 0.22	0.37 ± 0.21	0.603	0.34 ± 0.15	0.38 ± 0.22	0.309

*: PAF vs. ESUS, †: ESUS with SCAF vs. ESUS wo (= without) SCAF. AF = atrial fibrillation, BMI = body mass index, CHA_2_DS_2_-VASc = acronym of [Cardiac failure, Hypertension, Age ≥75 (2 points), Diabetes mellitus, prior Stroke or transient ischemic attack (2 points), Vascular disease, Age 65–74, Sex category (female)], DM = diabetes mellitus, ECG = electrocardiography, ESUS = embolic stroke of undetermined source, HTN = hypertension, PAF = paroxysmal atrial fibrillation, PWD = P wave duration, RMS = root mean square, SCAF = subclinical atrial fibrillation, TIA = transient ischemic attack.

**Table 2 jcm-09-01134-t002:** Logistic regression analysis for predicting recurrent stroke in patients with ESUS.

Variables	Univariate Analysis			Multivariate Analysis		
	OR	95% CI	*p*-Value	OR	95% CI	*p*-Value
Age	1.018	0.977–1.061	0.385	1.091	0.870–1.368	0.453
Sex	0.693	0.271–1.775	0.445	0.485	0.110–2.486	0.708
BMI	1.069	0.923–1.237	0.374	1.709	0.903–1.289	0.405
CHF	1.683	0.167–17.02	0.659	1.007	0.907–1.119	0.893
HTN	1.487	0.502–4.405	0.474	1.145	0.989–1.315	0.386
DM	1.599	0.516–4.954	0.416	1.211	0.706–1.826	0.674
Vascular disease	1.278	0.327–4.990	0.724	0.653	0.086–1.180	0.447
CHA_2_DS_2_-VASc	1.285	0.892–1.850	0.178	1.239	0.804–1.819	0.274
SCAF detection	2.727	1.040–7.149	0.041	1.881	0.705–5.019	0.207
PWD (≥135 ms)	3.029	1.244–7.377	0.015	2.756	1.061–7.161	0.037

BMI = body mass index, CHA_2_DS_2_-VASc = acronym of [Cardiac failure, Hypertension, Age ≥75 (2 points), Diabetes mellitus, prior Stroke or transient ischemic attack (2 points), Vascular disease, Age 65–74, Sex category (female)], CHF = congestive heart failure, CI = confidence interval, DM = diabetes mellitus, HTN = hypertension, OR = odds ratio, PWD = P wave duration, SCAF = subclinical atrial fibrillation.

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
