# Peer review of "Usefulness of P Wave Duration in Embolic Stroke of Undetermined Source"

_jcm, 2020, doi:10.3390/jcm9041134_

Round 1

Reviewer 1 Report

General comment:

This is in interesting report that appears to be well conducted, showing the use of SAECG for detecting PWD in patients with ESUS, identifying a cut-off at >135ms. This cut-off subsequently predicts stroke recurrence quite robustly, actually even more so than detection of AF (Table 2). If I am not mistaken, the latter should be stressed, and added into the Abstract. The report is generally well-written but there are a number of arguments that need clarification, and I have a list of minor points. I think the novelty is rather high.

Minor

Abstract.

Wording: “Stroke recurrence occurred in 22 patients (17.6%) and was significantly associated with PWD, not SCAF (odds ratio 2.756, 95% CI 1.061–7.161, p = 0.037).” I don’t understand what PWD, not SCAF refers to. Is it PWD but no simultaneous SCAF, or is it something else? What does the OR actually reflect?

Wording: “PWD, a biomarker associated with atrial substrate that directly contributes to AF and ESUS, is useful not only for predicting SCAF but also for identifying patients with atrial cardiopathy who have been overlooked due to overestimated SCAF burden.” I can understand what you try to state, but the argument of the second part of the sentence, does automatically follow from the first part. Please rephrase, perhaps define cardiopathy in the abstract? Or omit?

Introduction.

Please expand 1-2 sentences on the definition of “abnormal trigger” and “atrial substrate”, which may not be widely known to stroke physicians. Although these concepts are explained in the Discussion, but deserve some more mention in the Introduction.

Materials and methods.

Please state in what country the Chung-Ang Hospital is situated (Republic of Korea?).

P3, lines 109-110. It is unclear from the text whether all the follow-ups included Holter-ECG. Please state this more clearly.

Results

P3, line 141. …and 10 patients… please state “and 10 (8%) patients”, in accordance with the previous presentation.

Table 1. Please mention the abbreviation for SCAF (for consistency of mentioning all abbreviations).

Figure 1. Please give some more details in the Figure legend, referring to the Statistics section, what c-index represents. Is the same as convergence statistics (c-statistics) or something else? Please state “Cut-off: 135 ms”.

Figure 2. A hazard ratio (HR) with confidence intervals, should be possible to provide. Please add this in the Figure or in the text.

Figure 3. Would it be possible to add any statistics in the Figure, comparing the different relative ratios of stroke occurrence?

Table 2. “for the outcomes of recurrent stroke.” Please omit “the outcome of” as that makes a stroke scientist think of functional outcome.

P 6, lines 183-185. “Logistic regression analysis demonstrated that prolongation of PWD was a significant predictor of stroke recurrence, Table 2, odds ratio = 2.756, 95 % CI 1.061–7.161, p = 0.037” should not repeat all the numbers that is found in Table 2. Rather it should state “Logistic regression analysis demonstrated that prolongation of PWD was a statistically significant predictor of stroke recurrence (OR 2.8), more so than SCAF detection per se (OR 1.9, not significant)”.

Figure 4. Would it be possible to present some sort of statistical evaluation in the Figure or in the legend?

Discussion.

Top. The authors should conclude that PWD detection (above 135ms) predicts stroke recurrence more robustly than SCAF detection per se. This is what is shown in Table 2, and that’s quite interesting.

P8, lines 264-266 “Although the number of patients with stroke recurrence was small and statistical comparisons were not applicable, similar findings were observed in the present study when examining stroke recurrence (Figure 3).” This sentence is unclear. Are you referring to previous studies that could not be assessed statistically? As of now I get the impression, that your data set is too small for statistical analysis of these parameters? If so, the reason for not assessing certain fractions should be mentioned in the statistics section (statistical plan).

Section 4.3 has a quite abrupt ending. Please rephrase some sort of conclusion.

Limitations: “Clinical follow-up of twice a year after 1 year is not enough to support our conclusion”. Your conclusion has been to state that there are significant associations with stroke recurrence for one year – and your data supports that conclusion. The limitation rather deals with unknown associations beyond that time. Please rephrase.

Conclusion: Please add that the present data indicate that PWD above 135 ms, may be a more robust predictor of stroke recurrence than detection of AF.

Additional.

Section 4.3 first paragraph: You can probably shorten this quite a bit.

Your definition of AF lasting longer than 30s, complies with the accepted definition of AF, but there have been different definitions, see the meta-analysis by Mahajan R et al. Eur Heart J. 2018. PMID: 29340587. I think this possibility should be mentioned in the Discussion – i.e. PWD > 135ms may be associated to shorter durations of AF than 30s, thereby explaining an apparently stronger association with stroke, than AF defined as > 30s.

Author Response

Responses to Reviewer #1:

The authors thank the reviewer for the helpful and constructive comments.

This is in interesting report that appears to be well conducted, showing the use of SAECG for detecting PWD in patients with ESUS, identifying a cut-off at >135ms. This cut-off subsequently predicts stroke recurrence quite robustly, actually even more so than detection of AF (Table 2). If I am not mistaken, the latter should be stressed, and added into the Abstract. The report is generally well-written but there are a number of arguments that need clarification, and I have a list of minor points. I think the novelty is rather high.

Abstract.

Wording: “Stroke recurrence occurred in 22 patients (17.6%) and was significantly associated with PWD, not SCAF (odds ratio 2.756, 95% CI 1.061–7.161, p = 0.037).” I don’t understand what PWD, not SCAF refers to. Is it PWD but no simultaneous SCAF, or is it something else? What does the OR actually reflect?

            The authors thank the reviewer for pointing to this miscommunication. In our study, stroke recurrence in ESUS patients was significantly associated with prolonged P wave duration (PWD ≥135 ms on SAECG; OR 2.756, 95% CI 1.061–7.161, p = 0.037), however, SCAF detection per se was not related to stroke recurrence. This points is now clarified in the ‘Abstract’ as follows (page 1, line 30-32):

            …“Stroke recurrence occurred in 22 patients (17.6%), and prolonged PWD (≥135 ms) predicts stroke recurrence more robustly than SCAF detection per se.”

Wording: “PWD, a biomarker associated with atrial substrate that directly contributes to AF and ESUS, is useful not only for predicting SCAF but also for identifying patients with atrial cardiopathy who have been overlooked due to overestimated SCAF burden.” I can understand what you try to state, but the argument of the second part of the sentence, does automatically follow from the first part. Please rephrase, perhaps define cardiopathy in the abstract? Or omit?

            The authors thank the reviewer for the helpful and constructive comments. The authors fully agree with the reviewer’s opinion. As the reviewer mentioned, the sentence seems to be vague. Hence, the relevant sentence is now rephrased as follows (page 1, line 32-34):

            “In ESUS patients, PWD can be a useful biomarker to predict SCAF and further to identify subgroup patients who are more likely to have a recurrent embolic stroke associated with an atrial cardiopathy.”

Introduction.

Please expand 1-2 sentences on the definition of “abnormal trigger” and “atrial substrate”, which may not be widely known to stroke physicians. Although these concepts are explained in the Discussion, but deserve some more mention in the Introduction.

            The authors thank the reviewer for helpful and constructive comments. AF can be initiated by ectopic beats (abnormal trigger) and further can be maintained and progressed by atrial arrhythomogenic substrate defined as any structural or functional changes in atrial myocardium. This point is now clarified in the ‘Introduction’ section (page 2, line 53-55) as follows;

            “Generally, atrial arrhythmias can be initiated by ectopic beats (abnormal trigger) and further promoted by the atrial arrhythmogenic substrate which refers to any structural or functional change in atrial myocardium.” 

Materials and methods.

Please state in what country the Chung-Ang Hospital is situated (Republic of Korea?).

            Thank the reviewer’s careful comment. The Chung-Ang University Hosptial is located in Seoul, Republic of Korea. As you commented, the missing information for our institute is now provided in the ‘Matrials and Methods’ section (page 2, line 69).

P3, lines 109-110. It is unclear from the text whether all the follow-ups included Holter-ECG. Please state this more clearly.

            The authors thank the reviewer for pointing to this miscommunication. The patients in this study were followed up with a clinical examination, a 12-lead ECG and a Holter monitoring at 3, 6, and 12 months, and every 6 months thereafter. This point is now clarified in the ‘Matrials and Methods’ section (page 3, line 113).

Results

P3, line 141. …and 10 patients… please state “and 10 (8%) patients”, in accordance with the previous presentation.

            Thank the reviewer for the helpful comment. As recommended by the reviewer, the percentage values for each proportion of the ESUS patients were reported in the ‘Results’ section (page 4, line 146-147).

Table 1. Please mention the abbreviation for SCAF (for consistency of mentioning all abbreviations).

            Thank the reviewer for the meticulous review. As suggested by the reviewer, the abbreviation for SCAF were reported in the footnote of Table 1 (page 4, line 155).

Figure 1. Please give some more details in the Figure legend, referring to the Statistics section, what c-index represents. Is the same as convergence statistics (c-statistics) or something else? Please state “Cut-off: 135 ms”.

            The authors thank the reviewer for pointing to this miscommunication. As mentioned by reviewer, C-index is sometimes called the “concordance” statistic or C-statistics. The C-index is equal to the area under a ROC curve. Now, this point is clarified in the new Figure 1 and its legend. Also, “Cut-off: 135 ms” is stated in the new Figure 1 (page 5, line 165-167).

Figure 2. A hazard ratio (HR) with confidence intervals, should be possible to provide. Please add this in the Figure or in the text.

            Thank the reviewer for the helpful and constructive comment. As suggested by the reviewer, the statistic values for PWD (≥135 ms) is now added in the legend of Figure 2 (page 5, line 74-75).  

Figure 3. Would it be possible to add any statistics in the Figure, comparing the different relative ratios of stroke occurrence?

            Thank the reviewer for the important comment. As suggested by the reviewer, the authors add some statistics comparing the different relative ratios of stroke recurrence in the Figure 3, and now provide the new Figure 3 (page 6). Also, those statistic values are reported in the ‘Results’ section (page 6, line 189-194).

Table 2. “for the outcomes of recurrent stroke.” Please omit “the outcome of” as that makes a stroke scientist think of functional outcome.

            Thank the reviewer for the meticulous review. As recommended by the reviewer, we omitted the words “the outcome of” in the caption of Table 2, to avoid confusiong (page 7, line 207-208).

P 6, lines 183-185. “Logistic regression analysis demonstrated that prolongation of PWD was a significant predictor of stroke recurrence, Table 2, odds ratio = 2.756, 95 % CI 1.061–7.161, p = 0.037” should not repeat all the numbers that is found in Table 2. Rather it should state “Logistic regression analysis demonstrated that prolongation of PWD was a statistically significant predictor of stroke recurrence (OR 2.8), more so than SCAF detection per se (OR 1.9, not significant)”.

            The authors thank the reviewer for the constructive and thoughtful comment. As suggested by the reviewer, we rephrased the sentence which the reviewer mentioned (page 6, line 200-202) as follows;

            “Logistic regression analysis demonstrated that PWD prolongation was a more significant predictor of stroke recurrence (odds ratio [OR] 2.756), than SCAF detection per se (OR 1.881, not significant; Table 2).”

Figure 4. Would it be possible to present some sort of statistical evaluation in the Figure or in the legend?

            Thank the reviewer for the helpful and constructive comment. As suggested by the reviewer, the authors add the statistics comparing the stroke recurrences between patients with OAC use and patients with APT use in the Figure 4, and now provide the new Figure 4 (page 6-7). Also, that statistics value are reported in the ‘Results’ section (page 6, line 197-200).

Discussion.

Top. The authors should conclude that PWD detection (above 135ms) predicts stroke recurrence more robustly than SCAF detection per se. This is what is shown in Table 2, and that’s quite interesting.

            The authors thank the reviewer for the meticulous and constructive comment. Also, the authors fully agree with the reviewer’s opinion. As suggested by the reviewer, we added the sentence containing this interesting finding in the ‘Discussion’ section (page 7, line 220-221) as follows:

            …(4) prolonged PWD (≥135 ms) predicts stroke recurrence more robustly than SCAF detection per se

P8, lines 264-266 “Although the number of patients with stroke recurrence was small and statistical comparisons were not applicable, similar findings were observed in the present study when examining stroke recurrence (Figure 3).” This sentence is unclear. Are you referring to previous studies that could not be assessed statistically? As of now I get the impression, that your data set is too small for statistical analysis of these parameters? If so, the reason for not assessing certain fractions should be mentioned in the statistics section (statistical plan).

            The authors thank the reviewer for pointing to this miscommunication. Actually, the author tried to show that our results had a some similarity with that of previous study. Now, the authors rephrase the relevant sentence to avoid confusion (page 8, line 285-287) as follows;

            “Similarly, the present study also failed to show the superiority of OAC to aspirin in the secondary prevention of stroke for ESUS patients (Figure 4). However, when examining stroke recurrence patterns in this study, the different results may be expected (Figure 3).”

Section 4.3 has a quite abrupt ending. Please rephrase some sort of conclusion.

            The authors thank the reviewer for the meticulous and constructive comment. As suggested by the reviewer, the authors rephrased and added some sentences in ‘Section 4.3’ (page 9, line 293-297) as follows;

            “Therefore, considering the stroke recurrence trend shown in the present study, the current APT-based strategy is inadequate in preventing stroke recurrence in patients with ESUS. Further, our results suggest that the PWD can be a useful biomarker to identify patients who can benefit from OAC therapy.”

Limitations: “Clinical follow-up of twice a year after 1 year is not enough to support our conclusion”. Your conclusion has been to state that there are significant associations with stroke recurrence for one year – and your data supports that conclusion. The limitation rather deals with unknown associations beyond that time. Please rephrase.

            The authors thank the reviewer for the meticulous and constructive comments. As recommended by the reviewer, the sentence containing study limitations is now rephrase as follows (page 9, line 312-314);

            “The limitations of this prospective analysis study are the relatively short follow-up periods and the small number of patients who had a recurrent stroke event. However, these did not negate the ability of PWD to predict both SCAF and stroke recurrence in patients with ESUS.”

Conclusion: Please add that the present data indicate that PWD above 135 ms, may be a more robust predictor of stroke recurrence than detection of AF.

            The authors thank the reviewer for the helpful and constructive comment. Also, the authors fully agree with the reviewer’s opinion. As suggested by the reviewer, we added the sentence containing this essential finding in the ‘Conclusion’ section as follows (page 9, line 324-326):

            …” Furthermore, the present study indicates that advanced atrial substrate represented by PWD prolongation may be a more robust predictor of stroke recurrence than SCAF detection.”

Additional.

Section 4.3 first paragraph: You can probably shorten this quite a bit.

            The authors thank the reviewer for the helpful comment. As recommended by the reviewer, the authors shortened the relevant sentence which the reviewer mentioned (page 8, line 278-280).

Your definition of AF lasting longer than 30s, complies with the accepted definition of AF, but there have been different definitions, see the meta-analysis by Mahajan R et al. Eur Heart J. 2018. PMID: 29340587. I think this possibility should be mentioned in the Discussion – i.e. PWD > 135ms may be associated to shorter durations of AF than 30s, thereby explaining an apparently stronger association with stroke, than AF defined as > 30s.

            The authors thank the reviewer for the very helpful and insightful comments. The authors also completely agree with the reviewer’s opinion. As suggested by the reviewer, we discirbed the possible mechanism explaning the predictive value of PWD for recurrent stroke in ESUS patients in the ‘Disscussion’ section, with the reference literature (page 9, line 315-321) as follows;

            “Although this complies with the accepted definition of AF in general, there have been different definitions of AF in many clinical studies. A recent meta-analysis reported that SCAF strongly predicts clinical AF and is associated with elevated absolute stroke risk [30]. Concerning these findings, PWD ≥135 without SCAF detection may be related to AC with or without AF of shorter duration (<30 s) or very rare frequency, thereby explaining the predictive value of PWD for recurrent stroke. Further research is needed to refine the association between PWD and stroke.”

Reviewer 2 Report

The study by Moonki Jung et al. analyzed Usefulness of P wave Duration in Embolic Stroke of Undetermined Source. It is a case-control study. The study was performed on 125 cases with stroke of undetermined source and 125 patients with paroxysmal atrial fibrillation as controls. The authors adopted the PWD as a method for obtaining a precise and reproducible measurement of existing atrial substrate. The number of individuals in analyzed groups is not impressive but the study is of scientific importance.

I have several comments:

  1. Abstract: PWD abbreviation should be explained.
  2. Page 3, line 109, Follow-up Methods: How many follow-up visits were planned in general? According to information from Methods section there were at least four follow-up visits: at 3, 6, and 12 months after discharge, and every 12 months thereafter. However, according to Results section mean follow-up time was 1.6 years. Does it mean there were only 3 follow-up visits in some of the patients?.
  3. Page 3, Statistical Analysis Methods: Have you performed multivariate logistic regression analysis?
  4. Page 3, line 136: what do you mean by most of the tests? explain the exact number of P wave SAECG tests performed within 20–30 minutes. What about the rest of them? this should have been validated.
  5. What was the mean time of stroke recurrence?
  6. Have you analyzed the impact of other risk factors for stroke recurrence, e.g. DM? DM or hypertension were found to be risk factors for recurrent stroke. P wave duration should be analyzed together with the the confounding factors (multivariate logistic regression analysis).
  7. What is the diagnostic value of CHA2DS2-VASc score in your research? You should explain this analysis in the Methods section.

Author Response

Responses to Reviewer #2:

The authors thank the reviewer for the helpful and constructive comments.

The study by Moonki Jung et al. analyzed Usefulness of P wave Duration in Embolic Stroke of Undetermined Source. It is a case-control study. The study was performed on 125 cases with stroke of undetermined source and 125 patients with paroxysmal atrial fibrillation as controls. The authors adopted the PWD as a method for obtaining a precise and reproducible measurement of existing atrial substrate. The number of individuals in analyzed groups is not impressive but the study is of scientific importance.

I have several comments:

  1. Abstract: PWD abbreviation should be explained.

            The authors thank the reviewer’s meticulous comment. As recommended by the reviewer, the abbreviation of PWD is now provided in the ‘Abstract’ section (page 1, line 29-30).

  1. Page 3, line 109, Follow-up Methods: How many follow-up visits were planned in general? According to information from Methods section there were at least four follow-up visits: at 3, 6, and 12 months after discharge, and every 12 months thereafter. However, according to Results section mean follow-up time was 1.6 years. Does it mean there were only 3 follow-up visits in some of the patients?

            The authors thank the reviewer for the helpful and constructive comment. The follow-up interval from 12-months after discharge, was ever 6 months. The ‘12’ was a typo. The authors are very sorry for the reviewer. Also, during study periods, we recommended the patients to visit the clinic or emergency unit whenever they complained of AF-related symptoms, regardless of the scheduled follow-up visits. Please consider it with generosity. The typo was corrected (page 3, line 113). 

  1. Page 3, Statistical Analysis Methods: Have you performed multivariate logistic regression analysis?

            The authors thank the reviewer for the very important comment. The authors have performed a univariate and multivariate logistic regression analysis. Now, this point is clarified in the ‘Statistical Analysis’ section (page 3, line 123-127).

  1. Page 3, line 136: what do you mean by most of the tests? explain the exact number of P wave SAECG tests performed within 20–30 minutes. What about the rest of them? this should have been validated.

            The authors thank the reviewer for pointing to this miscommunication. Actually, all the P wave SAECG tests were completely finished within 30 minutes. This point is now clarified in the ‘Results’ section (page 4, line 145).

  1. What was the mean time of stroke recurrence?

            Thank the reviewer for the very helpful comment. The mean time of stroke recurrence in this study is approximately 252 days (mean ± SD: 252.0 ± 106.7 [day]). This missing value was reported in the ‘Results’ section (page 5, line 179-180). 

  1. Have you analyzed the impact of other risk factors for stroke recurrence, e.g. DM? DM or hypertension were found to be risk factors for recurrent stroke. P wave duration should be analyzed together with the the confounding factors (multivariate logistic regression analysis).

            Thank the reviewer for the very helpful and constructive comment. The authors also fully agree with the reviewer’s opinion. The authors have also performed a multivariate logistic regression analysis to identify the variables that were significantly related to a recurrent stroke event. This point is now clarified in the new Table 2 (page 7).

  1. What is the diagnostic value of CHA2DS2-VASc score in your research? You should explain this analysis in the Methods section.

            The authors thank the reviewer for the very important comment. As the reviewer know, CHA2DS2-VASc score are commonly used scheme for stratifying risk of stroke in patients with non-rheumatic atrial fibrillation (AF). Therefore, when selecting the control group with paroxysmal AF to compare with ESUS group, we excluded patients with a low risk of stroke (CHA2DS2-VASc score 0-1). This point is now reflected in the ‘Matrials and Methods’ section (page 2, line 88-89).